# Minstrel: Application-Aware SLM Inference Optimization on Edge Devices

Bakshree Mishra

*University of Illinois Urbana-Champaign*

Urbana, Illinois, USA

bmishra3@illinois.edu

*Abstract*—**Large language models (LLMs) have permeated different fields of computing, including agentic systems and controllers. Recent literature has introduced smaller language models (SLMs) capable of running on edge hardware, unlocking opportunities to significantly impact human and computer interaction. Following trends of LLM inference optimization on data centers, optimization of SLM inference on edge devices focuses on independently accelerating the prefill or decode phases. However, we expect the tasks targeted for SLM inference to not follow the same input and output length distributions as remote LLM inference, necessitating a reevaluation of options for hardware and software optimizations. Further, previous work does not study the impact of their optimizations in context of different downstream applications, and the benefits seen in their isolated evaluations are not generalizable.**

**In this work, we present Minstrel, an application-aware optimization framework for SLM inference on edge hardware. Minstrel introduces a hybrid empirical and analytical model to predict the inference latency for an application given an SLM and hardware. Using Minstrel, we divide the application space into a prefill-dominated P-Zone and a decode-dominated D-Zone. Leveraging the two zones, we make the observation that, for a certain range of applications, optimizing prefill phase is ineffective.**

*Index Terms*—**Edge inference, application-aware optimizations, small language models, SLM, Optimization zones**

## I. INTRODUCTION

Large language models (LLMs) and foundation models have become ubiquitous and entered the public consciousness with popular applications such as Chat-GPT [35] and Gemini [18], [46]. The generalizability of LLMs has influenced and reshaped how applications are developed in different fields; LLMs now are proposed for integration in operating systems [26], [36], as agents and controllers for sub-tasks in workloads [4], [14], [24], [31], [41], [50], [52]–[54], [59], [64], [70], and as a service (LLMaaS) [62], [63].

While LLM inference has typically required cloud resources such as large GPU clusters to run these large models, recent literature has introduced small language models (SLM) of parameter sizes 3B or lower [1], [25], [29], [45]–[47], [58], [68] that can be executed locally on edge/client devices such as smartphones and desktops with reasonable downstream accuracy. These show an encouraging trend in the ability to compress and distill information learned by the large language models [25], [47], [68] and can be a viable alternative for edge-inference tasks [19].

Prior work in accelerating LLM inference in data centers has leveraged the observation that different phases of LLM inference are memory and compute bound [67] to partition the phases into heterogeneous accelerators [9], [9], [37], [43], [58], [58], [60], [69]. However, the distribution of tasks for traditional LLMs and the new SLMs are different. First, a user is more likely to send larger texts (documents, images, etc.) as input to the larger models for complex tasks, and thus inference tasks can benefit immensely from accelerated prompt processing. SLMs are better suited for shorter prompts and for simple agentic tasks [20]. Further, SLMs with agentic use-cases can reuse shared prompts and context history [8], [15], [54], further reducing the prompt sizes to be processed. Other optimizations at data center level focus on improving inference throughput with batching and scheduling techniques [37], [65], [69]. It is unlikely that multiple inference requests may arise from the same user to leverage batching techniques on the edge. There is a need to systematically study the impact of different inference optimization techniques for SLM-oriented tasks with varying input and output lengths.

In this work, we propose Minstrel, a cross-stack framework that enables application-aware evaluation of hardware and/or software optimizations. Minstrel introduces a hybrid empirical and analytical performance model that predicts performance of an SLM on a hardware configuration. Minstrel learns the empirical model parameters for a hardware with linear regression over extensive offline profiling. Minstrel predicts whether a hardware and SLM can satisfy latency constraints for applications with different input and output lengths. Minstrel's performance model further analytically predicts the overall speedup achievable for applications with a wide range of prompt and output lengths, and identifies the inference phase to be prioritized for any hardware or software optimization. With Minstrel, we can clearly divide the inference application space, with varying prompt and decoded output lengths, into a P-Zone suitable for prefill acceleration and a D-Zone sensitive to decode optimization.

## II. BACKGROUND AND RELATED WORK

LLMs are deep neural networks that have had considerable success over natural language processing tasks as well as other modalities, primarily based on transformer [51] architecture.

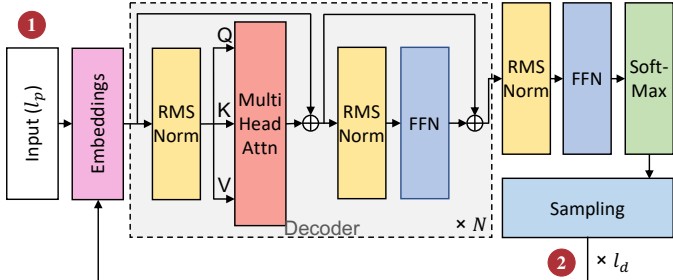

Fig. 1. Block diagram of a decoder-only transformer network. The decoder block consists of the multi-head attention and feed-forward network layers, and is replicated $N$ times in the network. $l_p$ : Prompt length, $l_d$ : Output length.

## A. Transformer Architecture

The baseline transformer architecture [51] consists of encoder and decoder blocks, and subsequent models derive from this architecture. Recent models such as GPT-3 [5] and Llama [48], [49] are decoder-only transformer models. Figure 1 shows a block diagram of a decoder-only network, consisting of components such as attention computation and feed forward networks. Inference with transformer architectures is autoregressive, where the generated embedding token is used to generate subsequent output token embeddings.

LLM inference can be decomposed into two phases. During the *prefill* phase (❶ in Figure 1), the model computes over the entire input prompt sequence (of length $l_p$). Inference performance with smaller sequence lengths is dominated by the feed-forward network in the transformer block, which scales linearly with sequence length; for longer sequence lengths, attention computation dominates which scales quadratically [32]. The KV (Key-Value) cache stores intermediate key and value projections during the prefill phase to avoid recomputation during decoding. Subsequently, the prefill phase generates the first output token. The *decode* phase (❷ in Figure 1) is autoregressive. Each iteration of the *decode* phase produces one token, which serves as an input for the next iteration to generate the next token. The *decode* phase continues depending on the length of output tokens ($l_g$). Thus, LLMs are stateful, where previously generated tokens provide context for subsequent output tokens.

## B. Heterogeneous Inference

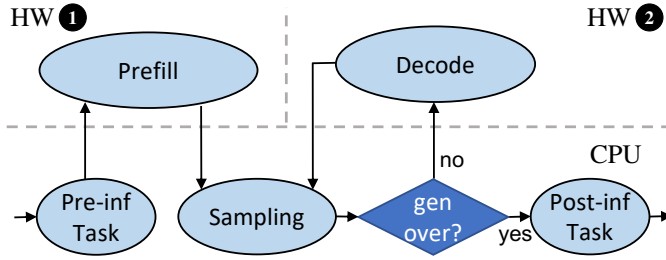

Fig. 2. Flow of executing LLM inference on heterogeneous hardware.

Figure 2 shows a flow chart for an LLM inference task using heterogeneous hardware (CPU, HW ❶ , and HW ❷

TABLE I
LLM INFERENCE ON HETEROGENEOUS HARDWARE.

| Work | HW ❶ | HW ❷ | Target |
|------|------|------|--------|
| Splitwise [60] | GPU (large) | GPU (small) | Datacenter |
| llm.npu [57] | NPU | CPU | Edge |
| HeteroLLM [9] | NPU | GPU | Edge |

). The CPU executes the upstream and downstream tasks. The LLM inference task primarily consists of prefill and decode phases, often offloaded to (same or different) hardware. In general, inference hardware can be CPU-only, or any combination of CPU, GPU, Neural Processing Unit (NPU), or other accelerators (e.g., [12]).

There has been extensive work in leveraging LLM model optimization techniques such as quantization [7], [55], FlashAttention [10], [11], KV caching [2], [40], inference under long context [56] and paged attention [22] to improve inference latency. Previous work [67] has further observed that prefill phase is compute intensive, while the decode phase is more memory intensive. Thus, LLMCompass [67] suggests that prefill phase (HW ❶ ) be mapped to hardware with more computation units, while the decode phase be mapped to hardware with less compute and more memory bandwidth (HW ❷ ). Previous research in data centers apply this strategy to map prompt prefill phase to larger accelerators [37], and further focus on scheduling and batching [37], [42], [44], [65] to improve inference throughput. Table I lists some of such recent work on heterogeneous LLM inference. Other inference scheduling research in data centers [34], [38], [61] optimize for delays and communication costs. In case of edge devices, recent work has attempted to apply similar partitioning of prefill and decode stages onto heterogeneous hardware available on these devices, with most leaning into optimizing prompt prefill.

## C. Application Characteristics of SLMs

Unlike LLM inference on cloud platforms, which support a variety of requests from multiple users, SLM inference on edge devices can be predictable depending on the application and use-case. One category of applications can be agentic use-cases [20] involving short incremental prompts and formatted outputs [4], [14], [24], [31], [41], [50], [52]–[54], [59], [64], [70], where the formatted outputs can be studied to predict the output/decode length. Edge inference of SLMs also provides the capability of caching and reusing context, usually a challenge for LLM inference in data centers [44]. Agentic applications, with multi-turn inference or common recurring prompt prefixes, can utilize this opportunity to reuse the KV-cache and reduce the effective prompt length for inference. However, current literature on optimizing SLM inference on edge devices propose solutions that do not consider these unique opportunities of SLM inference. For example, work on llm.npu (Table I) evaluates heterogeneous inference on benchmarks with long prompts and very short output lengths of 1–2 tokens, but does not study the performance with output lengths representative of actual tasks [39], nor does it consider

the impact of reusing the KV-cache from previous inferences that would effectively shorten prompt lengths. Further, other literature predicting LLM inference on hardware provide roofline estimations [21], [66] that map performance of LLM models as singular points, and do not study whether there is impact of any such application characteristics on the measured performance. One option to capture application-specific characteristics is to use the inference prompt and decode lengths as a proxy for complex application behaviors. For example, for an agentic task with common prompt prefixes, we can use the incremental prompt lengths during steady state to predict the actual inference computation. Similarly, we can use the (range of) output lengths observed within a task to predict the overall inference latency. Overall, SLM inference on edge devices provides more visibility to the range of applications and inference workloads. This provides an opportunity for more fine-grained evaluation of different optimizations and their impact per application, which is difficult to achieve with LLM inference on cloud platforms.

## III. MINSTREL

In this work, we introduce Minstrel, a framework to enable application-aware optimizations for SLM inference on the edge.

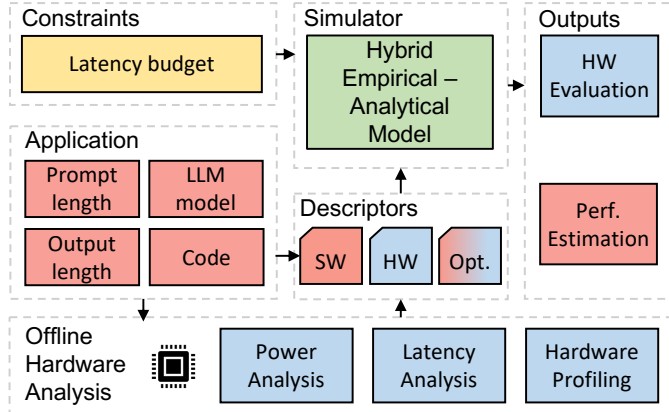

Fig. 3. A block diagram for the Minstrel system. Opt: Optimization.

### A. Hybrid Performance Model

Given a language model and hardware, we use a hybrid analytical and empirical performance modeling approach to estimate inference latency $\hat{t}_{e2e}$ for different prompt and decode lengths. SLM inference use-cases include tasks such as natural language commands and agentic tasks [20], [57], where sequence lengths are not very long. As a result, we can reason that the inference would be dominated by linearly-scaling feed-forward networks in the transformer blocks [32]. With this assumption, we define two linear models, $\hat{t}_{pp}$ and $\hat{t}_d$, that accept prompt and decode lengths as inputs and predict prefill latency and decode latency respectively as in Equations 1 and 2:

$$\hat{t}_{pp}(l_p) = \theta_0 l_p + \theta_1, \qquad (1)$$
$$\hat{t}_d(l_p, l_d) = \theta_2 l_p + \theta_3 l_d + \theta_4, \qquad (2)$$

where

$$l_p : \text{Prompt length,}$$
$$l_d : \text{Decode length,}$$
$$\hat{t}_{pp} : \text{Predicted prefill latency given } l_p,$$
$$\hat{t}_d : \text{Predicted decode latency given } l_p \text{ and } l_d,$$
$$\theta : \text{Learnable parameters.}$$

$\hat{t}_{pp}$ in Equation 1 is only a function of $l_p$, since prompt prefill phase always terminates with generating one output token and is independent of output length $l_d$. Minstrel assumes that the prefill and decode phases are sequential, and analytically expresses the end-to-end inference latency as in Equation 3.

$$\hat{t}_{e2e}(l_p, l_d) = \hat{t}_{pp}(l_p) + \hat{t}_d(l_p, l_d). \qquad (3)$$

We use empirical data obtained from running inference of SLMs on hardware to train the performance model. The linear parameters $\{\theta_0 \cdots \theta_4\}$ are learned by minimizing MSE.

### B. Hardware Evaluation with Minstrel

Minstrel uses Equation 3 to predict inference latency, given a model and hardware, for different applications. Hence, Minstrel can also evaluate whether the model and hardware can satisfy any latency constraint $t_{lim}$ set by the application with $l_p$ input prompt tokens and expecting $l_d$ output tokens. We can define `satisfy` to validate whether the model and hardware (system) can satisfy any latency constraint $t_{lim}$ set by the application with $l_p$ input prompt tokens and expecting $l_d$ output tokens as in Equation 4:

$$\texttt{satisfy}(l_p, l_d) = \begin{cases} 1, & \text{if } \hat{t}_{e2e}(l_p, l_d) \leq t_{lim}, \\ 0, & \text{if } \hat{t}_{e2e}(l_p, l_d) > t_{lim}. \end{cases} \qquad (4)$$

where

$$t_{lim} : \text{Latency constraint,}$$
$$\texttt{satisfy} : \text{System viability given constraint } t_{lim}.$$

Hardware evaluation with Minstrel can help application designers to preemptively test whether the application, with its prompt and decode lengths, can meet the set latency constraints. This can allow for application-level such as changing the prompt and/or decode lengths, or relaxing constraints, etc.

### C. System Optimization with Minstrel

Minstrel can evaluate impact of optimizing prefill and decode phases through hardware or software enhancements. Minstrel expresses the optimized inference latency $\hat{t}_{e2e}^{opt_{pp}, opt_d}$ as the sum of prefill and decode phases, each scaled by their respective optimization factors $opt_{pp}$ and $opt_d$.

$$\hat{t}_{e2e}^{opt_{pp}, opt_d}(l_p, l_d) = opt_{pp} \cdot \hat{t}_{pp}(l_p) + opt_d \cdot \hat{t}_d(l_p, l_d), \qquad (5)$$

$$\texttt{speedup}(l_p, l_d, opt_{pp}, opt_d) = \frac{\hat{t}_{e2e}(l_p, l_d)}{\hat{t}_{e2e}^{opt_{pp}, opt_d}(l_p, l_d)}, \qquad (6)$$

where

$$opt_{pp} < 1 : \text{Optimization factor for prefill stage,}$$
$$opt_d < 1 : \text{Optimization factor for decode stage,}$$
$$\hat{t}_{e2e}^{opt_{pp},opt_d} : \text{Predicted optimized latency,}$$
$$\texttt{speedup} : \text{Predicted speedup given } l_p, l_d, opt_{pp}, \text{ and } opt_d.$$

### D. Single Phase Optimization

We consider a special case where only one phase can be targeted for optimization. In such a case, the optimization factor for the unchanged phase can be assumed to be 1. Thus, for an application with prompt and decode lengths $l_p$ and $l_d$, Minstrel evaluates the maximum benefit, $\texttt{max\_speedup}$, that can be obtained through optimizing prefill or decode stages with factors as in Equation 7:

$$\texttt{max\_speedup}(l_p, l_d, opt_{pp}, opt_d) = \\ \max \begin{cases} \texttt{speedup}(l_p, l_d, opt_{pp}, 1), \\ \texttt{speedup}(l_p, l_d, 1, opt_d). \end{cases} \quad (7)$$

Equation 7 evaluates the impact of proposed optimizations with respect to individual applications. For an expected distribution of prompt and decode lengths in a system, Minstrel can evaluate optimization of which phase accelerates the inference more and enable prioritization of such optimizations.

## IV. METHODOLOGY

We evaluate Minstrel over 5 popular SLMs, Gemma 2 2b [17], Qwen1.5-1.8B-Chat [3], Phi 2 [30], Llama3.2 3B [28], [29], and Llama 3.2 1B Instruct [27], referred to as Gemma, Qwen, Phi, Llama3B, and Llama1B respectively. We use two consumer devices, an Intel i9-10900 CPU with an NVIDIA 3070 GPU [33] (referred to as *desktop*), and an NVIDIA Jetson Orin AGX as a substitute for a mobile device (referred to as *jetson*) [13]. We use the open source Llama.cpp [16] framework to measure inference latency. We profile over prompt lengths 1-1500 and generate outputs of lengths ranging from $\{0 \cdots 256, 512, 1024\}$ from the SLMs to create the performance dataset for the hybrid performance model. We choose this range for prompt lengths to match the range in llm.npu [57]. The Minstrel performance model uses linear regression to learn the learnable parameters $\{\theta_0 \cdots \theta_4\}$.

**Generalizing Inference with Synthetic Prompts**: While benchmarks with meaningful prompts are required to measure accuracy of a model's inference, the computation involved in an inference depends on the input length for computing prefill and filling the KV-cache, and on the output length for the number of autoregressive iterations. Hence, for an application, the effective prompt length (after considering any reuse of KV-cache from previous inference/common prompt prefixes) and the expected output length can be mapped to synthetic inference with the same prompt and decode lengths.

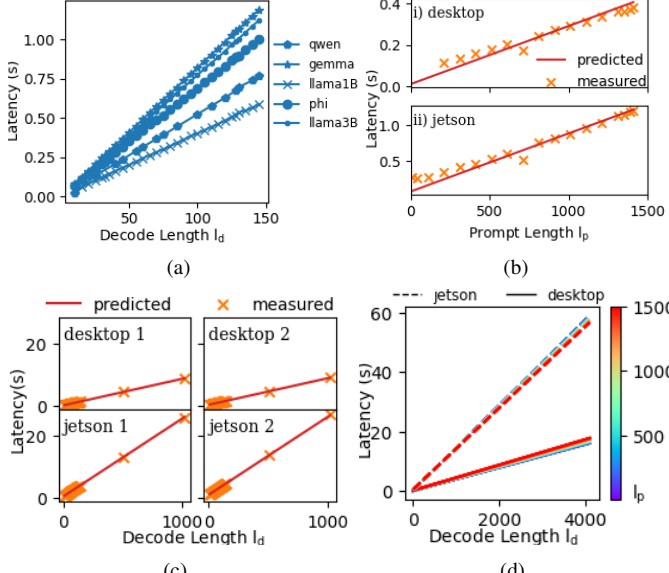

(a)                 (b)

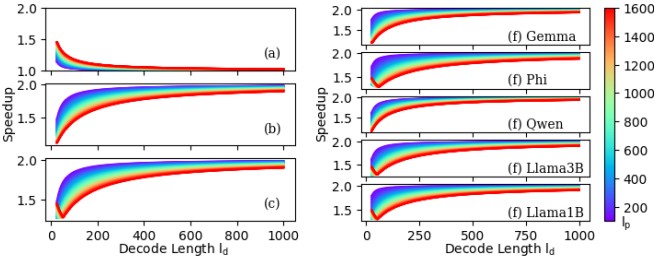

(c)                 (d)

Fig. 4. (a) Measured inference latency for different models with varying decode lengths for fixed prompt length of 32, (b) Linear increase in inference latency with increase in prompt length for Llama3B on *i) desktop* and *ii) jetson*, (c) Predicted and measured inference latencies for Llama3B on *desktop* and *jetson* for 2 prompt sizes, and (d) Predicted inference latency for a sweep of prompt and decode sizes with Llama1B on *desktop* and *jetson*.

Fig. 5. Y-axis shows predicted speedups of tasks with different decode lengths $l_d$ (on X-axis) and prompt sizes $l_p$ (denoted by the color-bar) by optimizing (a) only prefill phase and (b) only decode phase of Llama1B. (c) shows the max curve for (a) and (b) denoting the trade-off between optimizations. (d-h) show the optimization tradeoffs observed in 5 models.

## V. RESULTS

### A. Latency Analysis

Figure 4a shows the measured inference latencies for the 5 SLMs for a fixed prompt length and varying decode lengths $l_d$ on *desktop*. The X-axis denotes prompt length and Y-axis denotes inference latency in seconds. We observe that, for all the 5 models, the inference latency varies linearly with increase in decode length, as captured in Equation 2.

Figures 4b and c show the measured inference latency as well as Minstrel's latency prediction on both hardware configurations, *desktop* and *jetson*, for Llama3B. In Figure 4b, the output length is fixed (to 10) and prompt lengths are varied from 1 to 1500. We confirm linear scaling of latency with increase in prompt length, and that Minstrel is able to predict the expected latencies. In Figure 4c, desktop 1 and jetson 1 provide the measured and predicted inference latencies, on *desktop* and *jetson* respectively, for multiple decode lengths with fixed prompt size of 700; similarly, desktop 2 and jetson 2 provide the measured and predicted inference latencies

on *desktop* and *jetson* respectively for fixed prompt size of 1400. Once again, we observe that Minstrel's linear model reasonably captures the inference latency trends.

Finally, Figure 4d shows the predicted latencies from the performance model for Llama1B on *jetson* and *desktop*, for a sweep of prompt and decode lengths. The color-bar provides an indication of prompt lengths, with blue denoting shortest prompts and red denoting the longest. The solid lines show the predicted latencies for *desktop*, while the dashed lines denote the predicted latencies for jetson. We observe that the slope in *jetson* is higher than *desktop*. This indicates that the performance gap between *jetson* and *desktop* is narrow for short decode lengths, and widens with increase in decode length. With Figures 4a–d, Minstrel can test whether *jetson* and *desktop* can `satisfy` (Equation 4) an inference task with $t_{lim}$ set to, e.g., 5 seconds for different tasks. Thus, Minstrel can not only provide a tight latency estimation given an application's characteristics, but also guide the range of prompt and decode lengths feasible for an application targeting different hardware.

### B. Case Study: Single Phase Optimization

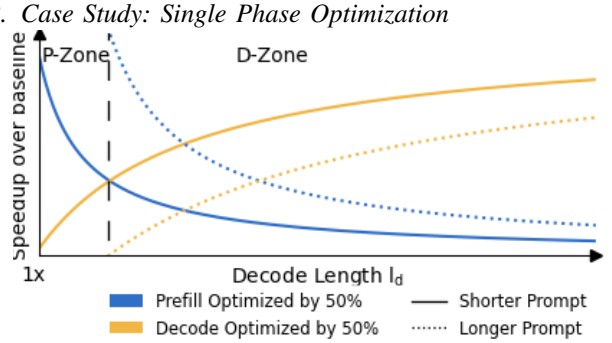

Fig. 6. Speedups trends on optimizing prompt and decode phases for different decode and prompt lengths. X-axis represents variation in decode lengths, solid and dotted lines represent short and long prompt lengths.

We use `speedup` and `max_speedup` from Equations 6 and 7 to predict the impact of independently optimizing prefill and decode phases for Llama1B in Figures 5a–c, where the X-axis denotes decode length $l_d$, and Y-axis denotes speedup observed over the baseline. For this evaluation, we choose $opt_{pp} = opt_d = 0.5$.

Figure 5a shows the Minstrel prediction for optimizing prefill phase by $opt_{pp}$. We observe that, for all prompt lengths, optimizing prefill phase sees diminishing returns with increase in decode length. Applications with small output lengths ($<$ 100) and/or long prompt lengths ($> 1500$) are noted to benefit from the prefill optimization. However, for decode length of $\geq 100$ tokens, we observe that the benefit of optimizing prefill phase diminishes for all observed prompt lengths.

Figure 5b shows Minstrel's prediction for optimizing decode phase by $opt_d$. We observe that applications with short prompt lengths ($< 1500$) benefit from optimizing decode phase even at short decode lengths. On the other hand, applications with longer prompts the benefit from the optimization at longer decode lengths ($\geq 100$ tokens).

Figure 5c plots the `max_speedup` from Equation 7 for prompt and decode lengths in Figures 5a and b. We observe

that Figure 5c first demonstrates a diminishing speedup trend corresponding to the speedup obtained from prefill optimization in Figure 5a, followed by a flattened trend corresponding to the speedup obtained from decode optimization in Figure 5b. We observe that all workloads, irrespective of prompt sizes, demonstrate similar trends of benefiting from prefill optimization for short decode lengths. Prefill optimization, although resulting in diminishing speedup with increase in decode length, remains the dominating optimization till a certain decode length ($\leq 100$ in Figure 5c). For longer decode lengths ($> 100$ in Figure 5c), we observe that the applications benefit more from decode optimizations.

Finally, Figures 5d-h summarize the `max_speedup` prediction trends for all 5 models evaluated, Gemma, Phi, Qwen, Llama3B, and Llama1B respectively (Figures 5c and h are identical). We observe that all models in Figures 5d-h demonstrate similar behavior, with a diminishing trend of benefits from prefill optimization for small decode lengths followed by a decode dominated range of decode lengths. Thus, depending on model, prompt, and decode lengths, an application would be better suited for optimization to different phases.

Figure 6 generalizes the trends in Figure 5 of optimizing just prefill or decode phases, shown in blue and yellow respectively. The X-axis denotes decode length $l_d$, and Y-axis denotes speedup observed over baseline. In general, the application-space with different prompt and decode lengths can be divided into two zones: a prefill dominated P-Zone, and a D-Zone where, primarily, optimization to decode phase impacts speedup. Determining the zone can be crucial to identify the dominating phase for optimization and enable systematic optimization efforts. By identifying these zones, Minstrel is the first step in capturing impact of application characteristics on SLM inference optimization on the edge.

### C. Discussion

Minstrel's linear model would not capture the quadratic scaling of attention computation for very large prompt lengths. However, for SLM inference use-cases covered by literature [20], [57], the sequence lengths are not very long. Thus for the purposes of predicting SLM inference latency, Minstrel provides reasonably good accuracy. Further, prior work has also noted similar linear increase in LLM inference latency with increase in prompt lengths [37].

The Minstrel model is limited to capture the impact of change to transformer blocks as an optimization factor. It may not be able to capture optimizations that are outside of the transformer blocks, such as the performance impact of speculative decoding [6], [23] with draft networks which can improve decode throughput. In such a case, however, Minstrel can still be used to provide upper and lower bounds for the benefits of speculation.

Energy is a significant factor for edge inference. Currently, Minstrel does not provide any insights into energy consumed by different configurations. For our future work, we plan to augment Minstrel to predict energy consumption of SLM inference.

## VI. Conclusion and Future Work

In this work, we present Minstrel, a framework to determine application-aware predictions of inference latency and the impact of optimizations. Minstrel introduces a hybrid empirical and analytical performance model for SLM inference on edge devices. Minstrel shows that for certain applications, naive acceleration of prefill phase has no impact on performance, and optimizing decode phase would be more valuable. Using Minstrel, we observe that the application space can be divided into P-Zone and D-Zone indicating prefill and decode dominated regions respectively and allow targeted optimization efforts for improving inference latency. Minstrel currently captures empirical hardware characteristics of a desktop and an NVIDIA Jetson device. In the future, we plan on extending our characterization to mobile SoCs. We also plan to validate Minstrel with hardware optimizations to improve prefill and decode phases. We also plan to introduce an energy-performance-offload tradeoff with Minstrel and increasing the solution space.

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
