# OpenReview forum: "Minstrel: Application-Aware SLM Inference Optimization on Edge Devices"
_iscaconf.org/ISCA/2025/Workshop/MLArchSys — MLArchSys 2025 Oral_

### Official Review · Reviewer_t3Uu · 2025-05-17
**Relevant idea, but the "application-aware" aspect is unclear throughout the paper**

**Confidence:** 3
**Rating:** 4

**Detailed Feedback And Questions For Authors:**

The paper addresses a timely and relevant area. Given the prevalence of edge applications and devices, there is a growing need to focus on developing smaller models such as SLMs for inference on these platforms.

However, there are two main weaknesses:

* While the hybrid modeling methodology appears sound, the modeling proposed in the paper seems overly simplistic. The paper does not fully address why such a simple approach using linear regression is sufficient to capture the nuances of SLM inference. Elaborating on the choice of the linear regression model, including a discussion or ablation studies on the characteristics of SLMs across different hardware architectures, would significantly strengthen the paper's claims.

* While the paper proposes an "application-aware framework," the evaluation section primarily focuses on prompt lengths and does not adequately address diverse SLM applications. Furthermore, it does not clearly explain how the framework achieves "application-awareness." Elaborating on the meaning of "application-aware" and detailing what unique aspects Minstrel addresses for each application beyond varying input and output lengths would greatly enhance the paper."

**Top Reasons To Accept The Paper:**

* The paper addresses a timely and relevant area. Given the prevalence of edge applications and devices, there is a growing need to focus on developing smaller models such as SLMs for inference on these platforms.

* The paper clearly discusses the differences between LLMs and SLMs, and why techniques used in prior work that focus on data-center-level optimizations may not be applicable in edge scenarios.

**Top Reasons To Reject The Paper:**

* While the hybrid modeling methodology is sound, the modeling proposed in the paper is overly simplistic. The paper does not fully address why this simplistic modeling with linear regression sufficiently captures the characteristics of SLM inference.
* While the paper proposes an 'application-awareness framework,' the evaluation section primarily focuses on prompt lengths and does not address diverse SLM applications, nor does it explain how the framework is 'application-aware.
*The validation of the model's effectiveness appears to rely on the fact that the evaluation is heavily memory-bound, leading to a linear increase in latency. However, it is unclear if this is true for all hardware or SLM models, which raises questions about the framework's modeling accuracy.

---

### Official Review · Reviewer_ZhZb · 2025-05-19
**Promising work for Application-Aware SLM Optimization on Edge Devices**

**Confidence:** 2
**Rating:** 6

**Detailed Feedback And Questions For Authors:**

- why is the assumption that the latency of SLMs is linear, is this too limiting?

- Is power important for edge computing?

**Top Reasons To Accept The Paper:**

- The paper introduces Minstrel, a tool that helps developers optimize  SLMs run on edge devices like phones, laptops.

- The paper is well motivated: Edge devices have different hardware than cloud servers.

- The too provides Speedup estimates and P-Zone / D-Zone Zone identification.

**Top Reasons To Reject The Paper:**

- Given that the motivation was edge devices, then energy use isn't really discussed.

- The test seems to be on synthetic prompts, not sure how will it generalise

---

### Official Review · Reviewer_37Gz · 2025-05-19
**This paper studies the optimization proposed for serving LLMs in the context of SLMs, and introduce Minstrel, a framework that predicts inference latency and the impact of such optimizations for SLMs.**

**Confidence:** 4
**Rating:** 7

**Detailed Feedback And Questions For Authors:**

Thank you for submitting your work to MLArchSys'25. I enjoyed reading your paper. The paper is well-written, it introduces the problem, and the background section helps with the context and preliminaries.

I may have missed this, but one question I have is, how does Minstrel deal with any new optimizations that get introduced very frequently for LLMs? It would be great to include a discussion about this in the final version of the paper.

Secondly, I didn’t quite understand this point made in the paper that "Edge inference of SLMs also provides the capability of caching and reusing context": I looked at the paper cited here, and what I gathered was that prompt-sharing is hard for LLMs when they are distributed across GPUs. So that is fine. How is this not a challenge, or how is this an opportunity, for SLMs in the context considered in the paper?

**Top Reasons To Accept The Paper:**

The paper is well-written, focuses on a specific problem in the context of SLMs, and provides results in a rigorous manner.

**Top Reasons To Reject The Paper:**

None.